# OpenReview forum: "Can Editing LLMs Inject Harm?"
_ICLR.cc/2025/Conference — Submitted to ICLR 2025_

### Official Review · Reviewer_uUvb · 2024-10-30

**Soundness:** 3
**Presentation:** 3
**Contribution:** 2
**Rating:** 6
**Confidence:** 3

**Summary:**

This paper reveals the risk that misinformation and bias can be injected into LLMs via knowledge editing.

**Strengths:**

1. The safety of LLMs is a very important issue.

2. The proposed method shows misinformation and bias can be easily injected into LLMs via knowledge editing.

3. There are many experiments to support the claim of this paper.

**Weaknesses:**

1. I wonder whether finetuning and ICL can be classified into knowledge editing. Is there any formal and widely accepted definition of knowledge editing?

2. Can the proposed method be applied to close-source models?

3. For open-source models, there are many existing works which show LLMs can be manipulated to general harmful information, such as Jailbreaking and malicious finetuning. What are the uniqueness of knowledge editing in achieving this goal?

4. Malicious finetuning has already been reported, such as Fine-tuning Aligned Language Models Compromises Safety, Even When Users Do Not Intend To!. Is it quite similar to the finding reported in this paper?

5. Can any kind of misinformation be formulated into structured format for ROME?

6. Is it better if more knowledge editing methods besides ROME are included in experiments?

7. The fact that ICL can mislead LLM to generate inappropriate information has been reported in some existing works such as "Many-shot jailbreaking". Is it similar with the ICL related finding in this paper?

**Questions:**

1. I wonder whether finetuning and ICL can be classified into knowledge editing. Is there any formal and widely accepted definition of knowledge editing?

2. Can the proposed method be applied to close-source models?

3. For open-source models, there are many existing works which show LLMs can be manipulated to general harmful information, such as Jailbreaking and malicious finetuning. What are the uniqueness of knowledge editing in achieving this goal?

4. Malicious finetuning has already been reported, such as Fine-tuning Aligned Language Models Compromises Safety, Even When Users Do Not Intend To!. Is it quite similar to the finding reported in this paper?

5. Can any kind of misinformation be formulated into structured format for ROME?

6. Is it better if more knowledge editing methods besides ROME are included in experiments?

7. The fact that ICL can mislead LLM to generate inappropriate information has been reported in some existing works such as "Many-shot jailbreaking". Is it similar with the ICL related finding in this paper?

---

### Official Review · Reviewer_MpRx · 2024-10-31

**Soundness:** 3
**Presentation:** 3
**Contribution:** 2
**Rating:** 5
**Confidence:** 3

**Summary:**

The paper proposes Editing Attack, a new type of safety risk for LLMs based on knowledge editing. The paper constructs a new dataset EditAttack and uses it to implement the attack with three knowledge editing methods: "Locate-then-Edit", fine-tuning, and in-context editing. The authors focus on two types of injection: Misinformation Injection and Bias Injection. Experiments show that the attack can inject commonsense misinformation and long-tail misinformation into LLMs, increasing the output bias, while keeping stealthy.

**Strengths:**

1. The proposed attack is effective while stealthy, demonstrating its potential to work in practice.
2. The experiments are well-designed. The metrics are sound and the findings are well supported.
3. The paper is clear and easy to follow.

**Weaknesses:**

1. The difference between the proposed attack and previous attacks such as jailbreaking and fine-tuning attacks is not convincing. Generally, all the previous attacks intend to manipulate the output of LLMs to include harmful, biased, or fake information, with methods like fine-tuning or in-context learning. Now the Editing Attack seems to fall into a subset of them, focusing on manipulating knowledge-related output.

2. There are no results for larger LLMs or black-box models. I fully understand the constraint of computation resources or budget consideration. However, larger models might be more resistant to manipulated knowledge and have stronger beliefs in their original knowledge, making it more difficult to implement the attack. For black-box models, it is possible to conduct experiments with fine-tuning and in-context editing.

3. As the attack is new, more discussions about the threat model, such as who are the attacker and victim, and the power of the attacker might be necessary to bring more practical implications.

**Questions:**

1. How could the attackers use in-context editing when implementing the attack in practice? Is this a kind of prompt injection attack?

**Details Of Ethics Concerns:**

The paper proposes a new attack that can inject misinformation and bias into LLMs.

---

### Official Review · Reviewer_vnRX · 2024-10-31

**Soundness:** 2
**Presentation:** 3
**Contribution:** 2
**Rating:** 5
**Confidence:** 5

**Summary:**

This work redefines knowledge editing as a novel type of safety threat to LLMs, Editing Attack, which includes subtypes misinformation and bias Injection. The authors introduce a new dataset, EDITATTACK, for attack implement and   evaluate the effects of Editing Attacks on LLM outputs。The authors also discuss the stealthiness of the attack and the challenges in defending against it,

**Strengths:**

It’s a new work that gave researchers a notice regarding Misinformation Injection and Bias Injection through knowledge editing to harm LLMs.
The authors provided implementation code.
The paper is  well-written with clear takeaways.

**Weaknesses:**

1. The motivation and practicality of editing attacks need further improvement. The authors present three types of attack injection methods: ROME, Fine-Tuning, and In-Context Editing. However, based on open-source LLMs, users typically do not directly use personally trained LLMs. The models that are widely used and would have a significant social impact are usually black boxes, making the attacks proposed by the authors unfeasible.

2. All the attacks are conducted on the original LLMs, such as Llama3. However, by introducing RAG, LLM is able to correct outputs based on external knowledge, which may render editing attacks ineffective, thus limiting the practicality of the proposed attacks.

3. Regarding Stealthiness, it's natural that a small amount of editing attacks are unlikely to have a significant impact on the general knowledge and reasoning capabilities of LLMs. A more reasonable experimental setup should adjust the proportion of editing attacks to cover the general knowledge of LLMs.

4. The paper provides the main findings through experiments, but the setup of the experiments could be improved to make the findings more insightful. For example, in Finding 1, it would be beneficial to provide the distribution of commonsense and long-tail misinformation to quantify it against the distribution of LLMs' known knowledge to reveal what commonsense shows and what long-tail shows. Moreover, it needs to be clarified how different degrees of deviation from common knowledge affect the effectiveness of the attacks.

5. For some experimental findings, it will be better to give reasons behind the phenomena for more insights. For example, in Finding 2, there needs to be more analysis and explanation of why bias editing attacks can affect the overall fairness of LLMs rather than only affecting the sensitive attribute targeted by the attack.

**Questions:**

Please elaborate further on the motivation behind editing attacks, and providing an example would be even better.

For widely used black-box LLMs, do attackers have practical approaches to carry out editing attacks?

By introducing RAG, which allows large models to correct erroneous inputs based on external knowledge, could this potentially render editing attacks ineffective?

How will Stealthiness change if editing attacks target the general knowledge of most LLMs rather than just a few?

How do we define what kind of information is commonsense and what is long-tail? Regarding the current definition of "long-tail misinformation injection (typically containing domain-specific terminologies, e.g., 'Osteoblasts impede myelination')", for a medical LLM, could it be argued that domain-specific terminologies do not count as long-tail?

Is there any assessment of misinformation injection effectiveness when misinformation with different degrees of long-tail level?

Why can bias editing attacks affect the overall fairness of LLMs rather than only affect the sensitive attribute targeted by the attack, as stated in Finding 2?

---

### Official Review · Reviewer_HFHC · 2024-11-03

**Soundness:** 3
**Presentation:** 2
**Contribution:** 2
**Rating:** 3
**Confidence:** 3

**Summary:**

The paper explores the risks of knowledge editing in Large Language Models, revealing that "Editing Attacks" can stealthily inject misinformation and bias into LLMs, impacting their fairness and accuracy.

**Strengths:**

1. The authors conduct a thorough investigation into the effectiveness of editing attacks on misinformation and bias injection, providing a comprehensive analysis of the risks involved.
2. The construction of the EDITATTACK dataset contributes to the field by offering a new resource for benchmarking LLMs against editing attacks, which can facilitate future research and development of defense mechanisms.

**Weaknesses:**

1. Although framing knowledge editing as a potential threat is helpful, its technical contribution is somewhat limited, and the results can be expected to not be entirely surprising.
2. The paper’s experiments focus on a few smaller LLMs (e.g., Llama3-8b, Mistral-v0.2-7b), limiting the findings' applicability to larger, state-of-the-art models that may respond differently to editing attacks. This narrow scope weakens the generalizability and robustness of the conclusions. For instance, ICE experiments could be conducted on more advanced models, such as Gemini or GPT-4o.
3. The proposed setting assumes that all injected editing knowledge consists of misinformation or bias, which may be impractical in real scenarios, as such content could be easily detected. It would be more insightful to examine results that mix misinformation or biased content with general, benign knowledge edits.

**Questions:**

N.A.

---

### Official Review · Reviewer_TjvK · 2024-11-04

**Soundness:** 3
**Presentation:** 1
**Contribution:** 2
**Rating:** 3
**Confidence:** 3

**Summary:**

The paper studies malicious knowledge editing in LLMs as a safety threat. In particular, it considers misinformation and bias. It finds both threats are very possible. Furthermore, commonsense misinformation is easier to inject than "long-tail" misinformation, i.e. highly domain-specialized misinfo. Bias injection can generalize to an overall biased model. And the paper finds evidence that these attacks can be hard to detect. Finally, the paper contributes a new dataset that can be used to study these types of attacks.

**Strengths:**

Important problem that may become even more salient as we increasingly rely on LLMs.

New dataset seems valuable to future research in this area.

While not too surprising (see below), the conclusions remain important.

**Weaknesses:**

My main concerns are with the writing and positioning of the paper. In particular:

The paper claims to "reformulate knowledge editing as a new type of threats for LLMs" (line 117-119, again 468-469). But this does not seem to be a new idea. I'm not an expert in knowledge editing, so please correct me if I'm mistaken, but my cursory search found a survey https://arxiv.org/pdf/2310.16218 that highlighted "if KME is maliciously applied to inject harmful knowledge into language models, the edited models can be easily transformed into tools for misinformation". https://arxiv.org/abs/2407.07791 seems to test this in a simulated setting. Similarly, https://arxiv.org/abs/2312.14302 and https://arxiv.org/abs/2408.02946 showed bias could be introduced and generalize beyond the training set, albeit with more geenric fine-tuning than knowledge editing. Furthermore, I'm also not sure about defining "its two emerging major risks" (line 119) as just misinformation injection and bias injection. For example, https://arxiv.org/pdf/2403.13355 suggests that backdoors could also be a knowledge editing threat. In general, it may be an overclaim that the problem framing is novel and comprehensive. Similarly, the "feasibility of disseminating misinformation" (line 123-124) already seems to have been demonstrated, and the "misuse risk of knowledge editing techniques" (line 122-123) in general. Significantly more precision seems needed to better highlight the contributions here vs what already exists in the literature, and avoid possibly substantial overclaims.

The extensive, ubiquitous usage of different formatting elements - bold, italics, etc. - with paragraphs containing text in as many as 4 styles, makes it harder to read. Suggest being much more judicious about what's really key to highlight.

Lines 461.5 to 465 seem redundant (those examples have already been explained), would cut.

Table 1 reports final numbers and the "performance increase", while table 2 reports original number, final number, and performance increase. The inconsistency seems a bit odd, and also the table isn't quite aligned in some places.

**Questions:**

In the evaluation, I was a bit unclear on what is actually calculating the scores - is it an LLM judging the responses or something else? Maybe worth a line in paper.

It seems the misinformation injection threat model is focused on a malicious actor doing it intentionally (line 50). If it's intentional, maybe calling it "disinformation injection" would be more appropriate (disinformation = intentional misinformation)? Could these threat models be a concern unintentionally?

Could you detect bias-edited model by doing a generic/standard bias eval? Especially if it makes the model generally biased and not narrowly so?

---

### Meta-Review · Area_Chair_CpKV · 2024-12-12

**Metareview:**

This work proposes the editing attack, which reveals the risk of knowledge editing in injecting harmful information into LLMs and affecting their fairness and accuracy. The EDITATTACK dataset it constructs and the findings it provides are recognized by the reviewers. However, the reviewers suggest that this work lacks obvious technical contributions, covers a limited number of LLM types, and has flaws in writing, indicating that it has not reached an acceptable state.

**Additional Comments On Reviewer Discussion:**

Reviewers generally believe that this work has obvious weaknesses. In particular, Reviewers HFHC, MpRx, and uUvb point out that the technical contribution of editing attack is limited and there is no significant difference with existing methods; Reviewers HFHC, vnRX, MpRx, and uUvb look forward to seeing the evaluation results of editing attack on larger LLMs and closed-source LLMs; Reviewer TjvK believes that the writing and positioning of the paper need to be improved. Given these weaknesses, the authors do not provide any rebuttal.

---

### Decision · Program_Chairs · 2025-01-22

Reject